# Binaural Pulse Modulation (BPM) as an Adjunctive Treatment for Anxiety: A Pilot Study

**DOI:** 10.3390/brainsci15020147

**Published:** 2025-01-31

**Authors:** Gerry Leisman, Joseph Wallach, Yanin Machado-Ferrer, Mauricio-Chinchilla Acosta, Abraham-Gérard Meyer, Robert Lebovits, Scott Donkin

**Affiliations:** 1Movement and Cognition Laboratory, Department of Physical Therapy, University of Haifa, Haifa 3498838, Israel; 2Resonance Therapeutics Laboratory, Department of Neurology, Universidad de Ciencias Médicas de la Habana, Havana 10600, Cuba; 3Maccabi Health System and Private Practice of Psychology, Modiin 7176538, Israel; drjosephwallach@gmail.com; 4Department of Neurology, Institute for Neurology and Neurosurgery, Havana 10400, Cuba; dra.yaninmachado@gmail.com (Y.M.-F.); chinchillamdneuro@gmail.com (M.-C.A.); 5Mental Wellness Society International, Beit Shemesh 9030860, Israel; meyer@bestmindsassociates.com (A.-G.M.); lebovits@hotmail.com (R.L.); 6Mind, Movement and Mood Wellness Centers, Movement Department, Lincoln, NE 68506, USA; scottwdonkin@gmail.com

**Keywords:** binaural pulse modulation, anxiety, neurofeedback, distress, fMRI, qEEG

## Abstract

Background: Treating psychiatric illnesses or influencing mental states with neurofeedback is challenging, likely due to the limited spatial specificity of EEG and the complications arising from the inadequate signal-to-noise ratio reduction of single-trial EEG. Objective: This pilot study aimed to investigate the feasibility of employing a binaural pulse mode-modulation (BPM) device to reduce anxiety by self-regulation. We desired to determine whether anxiety could be significantly reduced or regulated using BPM-type systems. Methods: Sixty adult participants were examined with self-reported anxiety tests (COVID Stress Scale, Generalized Anxiety Disorder 7, Beck Depression Inventory-II), which were completed before treatment, after four weeks, and after 12 weeks post-treatment. This BPM device produced two frequencies which combined to create a binaural pulse through differential auditory tone presentations. The participant calibrated the suitable target tone for optimal treatment efficacy. Each participant adjusted the binaural pulse to enhance the emotional intensity felt when envisioning an experience with comparable emotional significance or while performing a cognitive task while concurrently listening to music. The “treatment” relied on the individual’s regulation of binaural pulses to obtain the desired state. The training concentrated on particular facets of their psychological challenges while listening to an auditory tone, adjusting a knob until the sound amplified the intended emotional state. Another knob was turned to intensify the emotional state associated with distress reduction. Results: On the self-reported measures, the BPM treatment group was significantly better than the sham treatment (control) groups (*p* < 0.01). These findings indicate that over the four-week intervention period, BPM was similarly effective. On the GAD-7, the significant difference over time was noted before treatment and at the end of treatment for the experimental group, with the average GAD-7 score at the end of treatment being significantly lower (*p* < 0.01). Conclusions: BPM seems to induce a short-term alteration in self-reported distress levels during therapy. This study’s limitations are examined, and recommendations for future research are provided.

## 1. Introduction

### 1.1. Anxiety Disorders

Anxiety disorders manifest in various ways and are frequently persistent. Anxiety is a ubiquitous human experience. It denotes an emotional condition frequently characterized by distressing apprehension accompanied by numerous “emergency reactions”, such as palpitations or elevated heart rate. The Diagnostic and Statistical Manual of the American Psychiatric Association (DSM-5) differentiates between anxiety and fear, which may also intersect. Fear is a reaction to an actual or perceived immediate danger, while anxiety is the expectation of potential future harm.

Differentiating pathological anxiety from normative-range anxiety might be difficult. One must ascertain whether the anxiety arises from some medical condition such as hyperthyroidism, is a primary anxiety disorder (e.g., phobia), or is a manifestation of an underlying psychotic or mood-type problem. Complicating the situation, all three forms of anxiety may coexist in diverse combinations. Anxiety resulting from post-traumatic stress disorder may be intensified by thyroid illness. Therefore, meticulous consideration is required to elucidate the origins and temporal correlations of anxiety symptoms. The clinician must recognize that anxiety disorders frequently coexist with other problems. Diagnosis and differentiation frequently rely on a meticulous general medical and psychiatric history, as well as identifying “the types of situations that are feared or avoided and the content of the associated thoughts or beliefs”.

### 1.2. Binaural Beat Modulation 

Psychological therapies for anxiety disorders have often been less validated for their biological effects as opposed to their clinical outcomes. It is widely acknowledged that a deeper comprehension of brain alterations associated with effective psychotherapy may yield significant advantages. If we can discern the aberrant activation patterns associated with psychiatric symptoms, and if these patterns normalize post-intervention, we may utilize this information to build new treatment protocols aimed at the functional correlates of specific brain networks. This has already been proven in a clinical investigation [1]. Furthermore, we may be capable of directly targeting these diseased networks via neurofeedback or similar methods [1,2]. Decades of feedback studies utilizing electroencephalographic (EEG) signals have demonstrated that individuals can be trained to modulate the amplitude or topography of specific components of EEG activity [3]. Nevertheless, influencing particular mental states or addressing psychiatric diseases with EEG-based neurofeedback has proven challenging, perhaps due to its limited spatial specificity and the complications arising from the inadequate signal-to-noise ratio inherent in single trial-based EEG. Consequently, binaural sound modulation may provide a feasible option.

A function of the ear is transducing environmental stimuli into electrochemical potentials. The ear functions as a generator for the brain and nervous system [4]. The vestibule, a component of the ear, not only receives auditory information and transmits it to the brain but also converts bodily movements into energy [5]. 

Auditory stimulation conveys frequency data to the auditory cortex via the fourth layer of neurons, while beat, which is characterized by modulation, is transmitted to the auditory cortex by input modulation in the second and third layers [6]. The characteristics of the frequency–time structure of auditory signals is analogous to the neuronal frequency–time structure of impulsive flows and the anatomical basis of affective sound processing, suggesting that the mechanisms underlying the potential therapeutic effects of sound facilitate synchronization between afferent stimuli and endogenous neurodynamic processes, potentially influencing emotional states [7].

Binaural Beat Modulation (BPM) may be a promising novel method for modifying affective–cognitive function and the altering of emotional state. Auditory Beat Stimulation (ABS) in general and BPM can be of significant influence in a broad array of clinical applications. A comprehensive review of ABS and BPS is suggested for a deeper understanding of the possibilities of these technologies in mental health applications (cf. Chaieb and associates [8]). It has been suggested that related technologies such as ABS can be used to modulate cognition [9] to reduce anxiety levels [10], as well as to provide treatment for the effects of traumatic brain injury [11] and attention deficit hyperactivity disorder [12]. There have been mixed results reported in the literature concerning the appropriate auditory beat frequencies [13].

BPM can occur when either sine or square waves of closely related frequencies and stable amplitudes are presented binaurally simultaneously. For example, when a 440 Hz tone is presented to the right ear and a 414 Hz tone to the left, a beat of 26 Hz will be perceived, subjectively localized to the head of the participant. This effect was initially observed by H. W. Dove in 1839 and noticed again by Oster [14], who reported that B modulation could be perceived when there was a carrier frequency less than 1000 Hz. We can conclude from this early work that one requires a beat carrier frequency to be significantly low for cortical encoding. 

In attempting to employ BPM for anxiety reduction in those suffering from related disorders, be they trait or state types, Padmanabhan and associates [15] examined the effects of binaural beat audio on individuals manifesting pre-operative anxiety reactions. Measuring anxiety with the State–Trait Anxiety (STA-I) questionnaire, those patients having received binaural beats demonstrated a 26.3 reduction in the scores obtained on the STA-I when compared to the 11.8 reduction in STA-I scores in a placebo group. Weiland and associates [16] studied the effects of binaural beats on anxiety by providing natural sound with and without an embedded 10 Hz binaural beat. The STA-I scores in this study also demonstrated a significant reduction in anxiety levels in those individuals receiving binaural beats compared to those who did not. Le Scouarnec et al. [17] examined individuals suffering from anxiety disorders and also demonstrated a significant reduction in anxiety scores as compared to control patients not being exposed to binaural beats. Numerous similar effects have been found in the use of binaural beats in positively affecting mood states [18,19]. 

A similarity of frequencies between brain and musical rhythms is well known [1,20,21,22,23]. Low-frequency thalamocortical activity and musical rhythms have been documented to entrain [21]. The literature has extensively focused on the synchronization of neurodynamic processes and the physiological implications of this phenomenon. Synchronization events significantly influence the mechanisms of higher integrative brain activities [24]. This pertains to both neural activity induced by external input and also to endogenous neurodynamic processes. The formation of a conditioned response occurs at a specific level of synchronization between external stimuli: conditioned and unconditioned [25]. 

The coincidence of various activations of the temporal element is regarded as the paramount state for enduring alterations in synaptic efficacy [26,27,28]. An illustration of the significance of endogenic synchronization is the observation that attention and anticipated arbitrary motion coincide with synchronized neuronal discharges in the motor and nonspecific thalamus. The processes involved in the synchronization of brain activity are regarded as a significant mechanism of thalamocortical integration [29,30]. The synchronization of endogenic activity in the brain and nervous system with external stimuli is crucial for the brain, an organ that seeks information and stimulation [31,32]. There is much support for the idea that afferent impulses, along with specific stages of spontaneous neural activity, can result in a reorganization of the brain’s bioelectric activity. 

BPM can function as unconventional biofeedback, utilizing auditory tones to stimulate the nervous system. The sound is adjusted by each person to enhance the emotional intensity (negative or positive) felt while envisioning an experience and simultaneously listening to the device’s output. This pilot study aimed to investigate the feasibility of employing a BPM-type device to restore an optimal psycho-emotional state by stimulating endogenous self-regulation mechanisms to facilitate recovery from anxiety and mood disorders. We aimed to assess whether emotional distress would be modified (diminished) or managed by BPM-type systems.

## 2. Methods and Methodology

### 2.1. Participants

Sixty participants, who were randomly selected patients meeting the criteria specified in the next sections, from the outpatient clinics of the Institute for Neurology and Neurosurgery of Havana, participated in the study (28 males and 32 females aged between 20 and 78 years (M = 47.1 years s.d. 7.47)). All participants were diagnosed with anxiety and/or major depression and were assessed by a psychiatrist and/or psychologist according to the DSM-V criteria. None of the people assessed exhibited a history of neurological disease or dysfunction, trauma, or seizures, whether psychological or physical. All participants underwent physical tests that excluded typically assessed metabolic diseases such as hypothyroidism and any type of cancer. All participants were unmedicated, including corticosteroids and appetite suppressants. None of the patients examined exhibited or experienced potentially co-morbid diseases, such as asthma. The demographic and raw data are available on request at: (https://www.researchgate.net/publication/371935551_Binaural_Pulse_Modulation_BPM_as_Adjunctive_Treatment_of_AnxietyData, accessed on 29 January 2025).

### 2.2. Institutional Approval

The Institutional Review Board of the Institute for Neurology and Neurosurgery in Havana, Cuba, approved this project after careful review of informed consent and all ethical issues. The Institutional Review Board of the Institute for Neurology and Neurosurgery of Havana approval number is 2022-6, dated 7 February 2022. The file is available for inspection upon reasonable request to Dr. Yanin Machado-Ferrer (dr.yaninmachado@gmail.com).

### 2.3. Clinical Trial Registration

As the current investigation was a pilot study, FDA registration was not obtained at this point.

#### 2.3.1. Inclusion Criteria

Each participant presented or met the criteria for the diagnosis of an anxiety disorder, except for Post-Traumatic Stress Disorder (PTSD) which would have excluded a participant, and presented with a high level of distress associated with anxiety based on an objective screening measure. Participants could have been suffering from any one of seven types (i.e., generalized, panic, social anxiety disorder, separation anxiety disorder, or suffering from phobias). None of the individuals in the experimental group or control groups were taking any prescribed medication of any kind before and during the study. 

#### 2.3.2. Exclusion Criteria

None of the individuals examined presented with a primary psychiatric disorder or non-anxiety diagnosis, including developmental coordination disorder (DCD) or pervasive developmental disorder (PDD); active substance abuse or dependence excluding nicotine, caffeine, or cannabis; an active or inactive psychotic or thought disorder; hearing impairment; epilepsy or generalized seizure disorder; traumatic brain injury or cerebral palsy; a history of brain surgery; neurological abnormalities such as dyslexia; autism spectrum disorder (ASD); autoimmune disease; metabolic illnesses; a history of cancer; vascular disorder; or were currently breastfeeding.

### 2.4. Comparison Groups

The groups were the experimental (E) and control (C_s_), the latter of which received sham treatment. Sham treatment presented participants with white noise. Then, while they were exposed to the white noise, participants were directed to perform a cognitively taxing concentration task, such as reading or examining features in images, while being timed on the duration of their sustained focus. 

This occurred for up to 10-15 minutes. All participants had up to three sessions per week, for 12 weeks.

### 2.5. Randomization and Blinding

Randomization for participant allocation to the two groups was provided. Randomization was performed via a randomized block design with varying block sizes of two, four, or six participants. In each block, one-half of the participants were randomly assigned to Group C_s_, and the other half to Group E. Randomization was achieved using a randomized block design with block sizes of two, four, or six participants. Fifty percent of the participants in each block were randomly allocated to Group C_s_, while the remaining fifty percent were assigned to Group E. Randomization was achieved by computer-generated sequence technology, guaranteeing that both the randomization process and the allocation sequence remained disguised from investigators and participants. Concealment was guaranteed as outlined below: Every computer-generated randomization sequence was distinct and irreproducible. Randomization was implemented for Group E or Group C_s_. Only the designated individual at the study site knew which assignment corresponded to which experimental treatment or control group, with this information not revealed until study unblinding occurred and after all data had been entered into the database and the database sealed before statistical analysis. 

### 2.6. Procedure

#### 2.6.1. Anxiety Evaluation

Each participant was evaluated at the outset of the study, at the end of the four-week treatment phase, four weeks post-treatment phase, and twelve weeks post-treatment phase after the study. Participants completed the following standardized instruments: the Beck Depression Inventory-II (BDI-II) [33] or Geriatric Depression Scale, the Generalized Anxiety Disorder test (GAD-7) [34], and the COVID Stress Scale (CSS) [35].

#### 2.6.2. Apparatus 

Auditory stimulation was delivered by BPM within the frequency range of 0–350 Hz. The audio stimuli were calibrated in volume for each ear individually. The primary auditory frequency adjustment knob (frequency) regulated a range from 0 to 330 Hz, while the secondary auditory frequency adjustment knob (disruptor) managed an additional offset range from 0 to 20 Hz. The primary tone-specific level at a frequency control setting of 2 was approximated at 75 Hz, whereas the disruptor at the same setting generated an offset of around 20 Hz, inducing overtones in the beta region. This disparity is recognized and integrated in the brain as the binaural pulse. Bone conduction headphones were employed.

#### 2.6.3. BPM Administration

BPM is not a typical biofeedback device; using auditory frequencies, it activates the neural system. The sound is adjusted by each person to enhance the emotional intensity (positive or negative) felt while envisioning an experience, while simultaneously listening to the device’s audio. The administration is more fully described in the flow diagram represented in Figure 1.

### 2.7. Intervention Procedure with BPM

The participants became acquainted with the BPM device and were provided with an explanation of its usage and the independent volume controls for each ear. Participants were instructed on the configuration and modulation of the frequency and disruptor control knobs. The participants were thereafter directed to don the headphones and activate the device, setting both the frequency and disruptor to 2. They thereafter listened to the tone for several minutes to acclimate and familiarize themselves with it. They adjusted the volume and control knob until they found a frequency with which they were most comfortable. When this was achieved, they were told to turn the BPM off to hear the next step. Then, they were required to identify and describe a target relaxing or positive experience, thought, and associated image. Then, the participants were instructed to attempt to continue to engage in focusing on the target experience by continuing to think about it and the image, and then to turn on the BPM while they wore the headphones and heard the tone. The participants were instructed to adjust the frequency control knob until they felt a slight intensification of the feeling of relaxation. Then the participants continued to focus on the feeling while they slowly adjusted the disruptor control knob until they felt an even stronger increase in the feeling, or at least the feeling did not reduce. The participant was instructed to continue listening to the auditory stimuli for 15-20 minutes, with the option for a break followed by a second 15–20-minute treatment session. Treatment was twice per week for four weeks.

### 2.8. Statistical Analysis

A one-way ANOVA test for repeated measures was employed as it compares the means of three or more treatments involving the same group of individuals (or matched subjects) for each treatment. The tests were employed to analyze disparities between groups, temporal factors, and interactions for each of the measures to determine if self-reported emotional distress was significantly reduced in the BPM treatment group relative to the control group. Significance was set at the 0.01 level. 

Effect size was calculated, providing a standardized measure of the strength or magnitude of the effect. Cohen’s D gave us a standardized way of assessing the *magnitude* of the effect. We also employed Hedges’ g to reduce bias.

## 3. Results

Analysis of Variance tests were used to evaluate differences between groups, time, and interactions for each of the measures to determine if self-reported emotional distress was significantly reduced in the BPM treatment group relative to the control group. Reflected in Table 1 on the GAD-7, significant differences were present between groups (F 6.30, *p* < 0.01), over time (F = 8.75, *p* < 0.01), and between groups (F 2.99, *p* < 0.01), with the experimental group showing significant improvement. Table 2 presents the medium effect size between groups (partial eta 2 = 0.06), which for time was medium (partial eta 2 = 0.11), and for the interaction between time and group was medium (partial eta 2 = 0.11). 

Assessments of the fixed effect terms on the GAD-7 were conducted using F tests and represented in Table 1. The null hypothesis for the test is contingent upon whether it pertains to a fixed factor term or a covariate term. For a constant factor term, the null hypothesis posits that the term does not exert a substantial influence on the response. The null hypothesis for a covariate term posits that there is no association between the response variable and the covariate. Here we support a fixed factor term.

In Table 2, we measure the effect size on the GAD-7. Effect size here measured the strength of the relationship between the two variables time and group. It assisted in determining the practical significance of the findings, independent of sample size, and employing Cohen’s d.

In Table 3, we examine the significance of the difference before and at the end of the treatment. On the GAD-7, the significant difference over time was between the measures before treatment and at the end of treatment for the experimental group, with the average of the GAD-7 score at the end of treatment being lower (2.79, *p* < 0.01) and reflected in Table 3.

Table 4 provides the results of the interaction between groups over time, showing a statistically significant difference between the experimental treatment and control groups at the end of treatment, with the experimental treatment group having a lower mean GAD-7 score at the end of treatment. Additionally, the effect was limited, and at both 4 and 12 weeks post-intervention the results show a return to higher GAD-7 mean values.

On the GAD-7, the significant group difference was between the experimental treatment group and the control group, with the control group having a higher mean score (-2.16, *p* < 0.01). The effect was limited, as seen in Table 5, and at both 4 and 12 weeks post-intervention the results show a return to higher GAD-7 mean values. The clinical effects were robust but tended to reduce in the testing at 12 weeks post-intervention.

On the CSS, represented in Table 6, no significant differences were found between groups (F = 1.68, *p* > 0.05), a significant difference was present over time (F = 5.87, *p* < 0.01), and the groups appeared to behave similarly over time (F = 0.47, *p* > 0.05). The effect size for the difference over time was medium (partial eta 2 = 0.075) with differences between the initial screening as well as baseline and end of treatment and four weeks after the end of treatment. 

On the PCL-5, reflected in Table 7, no significant differences were present between groups (F 0.75, *p* > 0.5), and no significant differences were present over time (F = 0.91, *p* > 0.1); however, significant differences were present in the interaction between groups over time (F 2.34, *p* < 0.05). The effect size for the interaction between time and group was medium (partial eta 2 = 0.09). 

Table 8 presents the results of the interaction between groups over time, showing a statistically significant difference between the experimental treatment group to all the other groups at the end of treatment, with the experimental treatment group having a lower mean PCL-5 score at the end of treatment. 

Table 9 demonstrates that the effect was limited and at 12 weeks post-intervention the results show a return toward higher PCL-5 mean values. 

On the BDI-II, represented in Table 10, no significant differences were found between groups (F = 2.38, *p* > 0.05), over time (F = 1.54, *p* > 0.05), or in the interaction between groups and time (F = 0.42, *p* > 0.05).

## 4. Discussion 

Evidence exists supporting the presence of bidirectional and directional connectivities between the brainstem and the limbic and auditory systems. Retrograde tracing has revealed extensive and direct connections between the inferior colliculus [36] basal nucleus of the amygdala. Furthermore, findings from lesion studies suggest that activation of the inferior colliculus by emotionally unpleasant information is modulated by serotonergic inputs from amygdaloid central and basolateral nuclei [37]. Lesions in the thalamus and auditory midbrain have been demonstrated to inhibit conditioned autonomic emotional responses associated with auditory stimulation [38]. Research indicates that even in primates, there exists a widespread network of connections between the limbic and cortical auditory regions [39]. Research on rhesus monkeys indicates that auditory stimulation activates the amygdala; however, this activation does not occur if the inferior colliculus is ablated [40]. The efforts to functionally describe this route and its modulatory impact on real-time encoding have been notably diffuse, especially across various species [41,42,43]. Subcortical alterations have been evidenced at the initial phases of sensory and motor processing streams, further challenging the concept of passive sensory processing. Suga [44] has illustrated modulation to the furthest boundaries of the auditory periphery at the cochlear hair cell level. 

A burgeoning corpus of information illustrates the influence of supplementary factors on auditory processing. Subcortical alterations have been evidenced at the initial phases of the sensory and motor processing streams, further challenging the concept of passive sensory processing. Suga [44] has illustrated modulation to the furthest boundaries of the auditory periphery at the cochlear hair cell level. 

Support based on numerous methodological approaches has illustrated the influence of auditory processing on emotional states [38,45,46,47,48]. The existence of both direct and bidirectional pathways linking the limbic system with the auditory brainstem is confirmed by anatomical evidence. Retrograde tracing has demonstrated direct and broad connections between the inferior colliculus and the amygdala’s basal nucleus [49]. Moreover, lesion study results have shown that aversive information generated from the inferior colliculus is influenced by the amygdala’s basolateral and central nuclei [37]. Lesions in the thalamus and auditory midbrain have been reported to be able to suppress conditioned autonomic emotional responses to sound stimulation [38]. Studies demonstrate that monkeys have a network of connections linking the auditory periphery to limbic areas [39]. Auditory stimulation in rhesus monkeys activates the amygdala; however, inferior colliculus ablation inhibits this activation [40]. The attempts to functionally describe this pathway and its modulatory effects have been relatively fragmented, especially across different species.

An effective meta-analysis was performed [13] that analyzed 22 studies based on 35 effect sizes that demonstrated consistent significant effects on anxiety and pain reduction. Their meta-analysis contributed to supporting the notion that binaural-beat exposure is effective in reducing anxiety levels without prior training. While many studies have examined binaural pulse modulation, many up to now have reported contradictory or inconclusive results. Some studies consistently report a diminishing impact of BPM on anxiety over time, with the underlying mechanisms, how it is that the BPM is produced, and which cortical networks are most involved yet to be understood. Knowing the basis of the effect will support the BPM stimulation optimization as a potentially powerful therapeutic tool with the capacity to modulate cognitive and mood states [50]. We have already performed such a pilot study examining electrophysiological and fMRI changes as a consequence of BPM stimulation that is consistent with the psychometric findings [1]. Additional research with more precise reporting of research methodologies, in particular including studies performed in clinical environments, will aid in the clarification of BPM effects on anxiety, mood, and other behavioral aberrations. Numerous considerations may impact the efficacy of BPM, including the duration of the implied stimulus carrier, frequencies chosen, and background noise that could potentially impact the results. Frequencies may also play a role as well as the addition of background, white, or pink noise, which may amplify the beat frequency, and having already been subjectively noted to vary the results, with a more robust effect noted at 432 Hz rather than 440 Hz.

A study on aging effects revealed that a gamma range BPM in the EEG could be identified regardless of age, although older participants exhibited reduced accuracy in detection [51]. Certain studies have indicated gender disparities in BPM perception and changes in auditory perception throughout the menstrual cycle [52]; however, such variations have been dismissed under typical conditions [53]. Other research indicates that focusing on the stimuli [54] may influence the effectiveness of BPM, as several extra variables may come into play. Electrophysiological studies examining the effects of auditory beats across various stimulation conditions and parameters remain scarce. Such investigations are essential for formulating appropriate hypotheses that elucidate the clinical and behavioral consequences of BPM.

### Study Limitations

The study was designed as a pilot investigation justified based on clinical data that had been recently published [1]. Numerous limitations exist in being able to effectively conclude robust clinical effects. We have observed significant short-term effects of BPM over multiple testing iterations, but long-term follow-up studies have not been performed. Additionally, we have not studied the optimization of frequency parameters for BPM and have not evaluated BPM’s applicability across demographics, as the study was performed in Havana, Cuba. We have also not differentiated between the numerous potential sources of anxiety and the comorbidities of anxiety. The study could have incorporated additional tests to strengthen the reliability and validity of the findings—even if the results were robust. Additional tests might have included the Hamilton Anxiety Rating Scale (HAM-A) for anxiety assessment and the State–Trait Anxiety Inventory (STAI) to differentiate between state and trait anxiety. Physiological measures, such as heart rate variability, could have provided an objective corroboration of self-reported data. This study is a pilot to justify a larger funded and registered clinical trial.

## 5. Conclusions

The results of the BPM treatment group were similar to the treatment-as-usual group, which included either psychiatric medication or psychotherapy. Furthermore, on the self-reported measures (GAD-7, CSS), the BPM treatment group was statistically better than the sham treatment and the waitlist control groups. These findings indicate that over the four-week intervention period, the BPM, as used in this study, was similarly effective to the standard treatment approach for anxiety. Future controlled clinical trials consisting of self-reported inventories and electrophysiological studies including heart rate variability will need to be performed to more fully realize the potential therapeutic benefits in treating anxiety and other related disorders, perhaps even PTSD. Due to the self-directed nature of this treatment approach and the beneficial results without the costs and side effects from medication or psychotherapy, BPM intervention appears to provide a potentially significant tool in the ongoing treatment of anxiety. With the increased presence of psychiatric and psychological complaints, the potential benefits of this intervention as an adjunctive therapeutic tool may be profound. Additionally, with concerns of medication side effects, in the short- and long-term, this intervention may provide a benefit in reducing side-effect severity and perhaps even reducing reliance on long-term medication use for anxiety.


## Figures and Tables

**Figure 1 brainsci-15-00147-f001:**
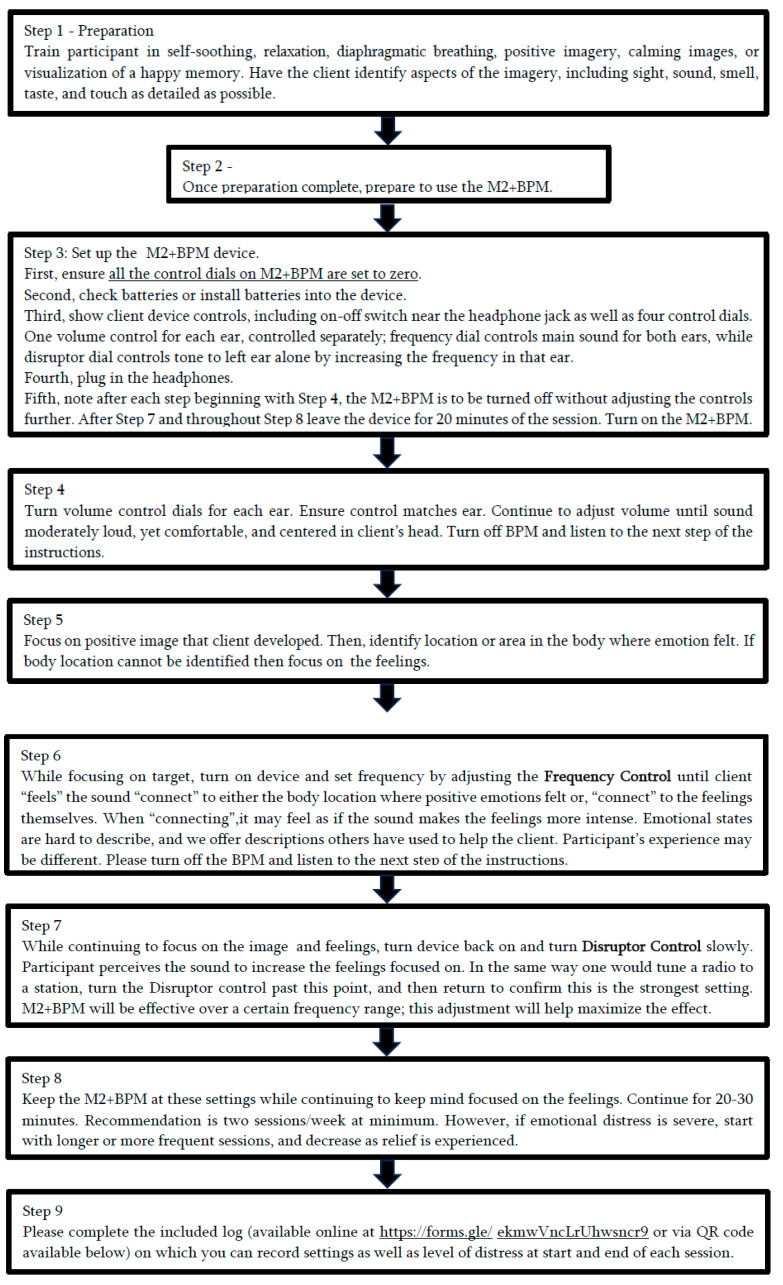
Flow chart of BPM Administration.

**Table 1 brainsci-15-00147-t001:** Type 3 tests of the fixed effects on the GAD-7.

Effect	Num DF	Den DF	F Value	Pr > F
Group	3	15	6.30	0.0056
time	4	60	8.75	<0.0001
time×Group	12	60	2.99	0.0025

**Table 2 brainsci-15-00147-t002:** Effect size on the GAD-7.

Obs	Effect	Partial_eta_2
1	Group	0.06077
2	time	0.10733
3	time * Group	0.11237

**Table 3 brainsci-15-00147-t003:** GAD-7 differences prior to and at the end of treatment.

Obs	Effect	Time	Group	_time	_Group	Estimate	StdErr	DF	tValue	Probt	Adjustment	Adjp
1	Group	_	A	_	B	−0.6900	0.5412	15	−1.28	0.2217	SMM	0.7404
2	Group	_	A	_	C	−1.2000	0.5102	15	−2.35	0.0328	SMM	0.1669
3	Group	_	A	_	D	−2.1600	0.5102	15	−4.23	0.0007	SMM	0.0042
4	Group	_	B	_	C	−0.5100	0.5412	15	−0.94	0.3609	SMM	0.9125
5	Group	_	B	_	D	−1.4700	0.5412	15	−2.72	0.0159	SMM	0.0854
6	Group	_	C	_	D	−0.9600	0.5102	15	−1.88	0.0794	SMM	0.3604

**Table 4 brainsci-15-00147-t004:** Interaction between the experimental treatment group compared to all the other groups at the end of treatment, with the experimental treatment group having a lower mean GAD-7 score at the end of treatment.

Obs	Effect	Time	Group	_time	_Group	Estimate	StdErr	DF	tValue	Probt	Adjustment	Adjp
1	time	BL		END		2.7875	0.5880	60	4.74	<0.0001	SMM	0.0001
2	time	BL		END + 12		−0.1375	0.5880	60	−0.23	0.8159	SMM	1.0000
3	time	BL		END + 4		0.8000	0.5880	60	1.36	0.1787	SMM	0.8460
4	time	BL		PRE		3.13 × 10^−14^	0.5880	60	0.00	1.0000	SMM	1.0000
5	time	END		END + 12		−2.9250	0.5880	60	−4.97	<0.0001	SMM	<0.0001
6	time	END		END + 4		−1.9875	0.5880	60	−3.38	0.0013	SMM	0.0126
7	time	END		PRE		−2.7875	0.5880	60	−4.74	<0.0001	SMM	0.0001
8	time	END + 4		PRE		−0.8000	0.5880	60	−1.36	0.1787	SMM	0.8460
9	time	END + 12		END + 4		0.9375	0.5880	60	1.59	0.1161	SMM	0.6908
10	time	END + 12		PRE		0.1375	0.5880	60	0.23	0.8159	SMM	1.0000

**Table 5 brainsci-15-00147-t005:** Limited effect size at both 4 and 12 weeks post-intervention. Results demonstrate a return to higher GAD-7 mean values.

Effect	Time	Group	_time	_Group	Estimate	StdErr	DF	tValue	Probt
time * Group	END	A	END	B	−4.5000	1.2101	60	−3.72	0.0004
time * Group	END	A	END	C	−5.0000	1.1409	60	−4.38	<0.0001
time * Group	END	A	END	D	−7.6000	1.1409	60	−6.66	<0.0001
time * Group	END	A	END + 12	A	−7.2000	1.1409	60	−6.31	<0.0001
time * Group	END	A	END + 4	A	−4.8000	1.1409	60	−4.21	<0.0001
time * Group	END	A	PRE	A	−7.4000	1.1409	60	−6.49	<0.0001

**Table 6 brainsci-15-00147-t006:** CSS test data between groups.

Obs	Effect	Time	Group	_time	_Group	Estimate	StdErr	DF	tValue	Probt	Adjustment	Adjp
1	time	BL		END		14.3000	4.3809	65	3.26	0.0018	SMM	0.0172
3	time	BL		END + 4		14.3833	4.3809	65	3.28	0.0017	SMM	0.0163
7	time	END		PRE		−15.4208	4.3809	65	−3.52	0.0008	SMM	0.0079
8	time	END + 4		PRE		−15.5042	4.3809	65	−3.54	0.0007	SMM	0.0074

**Table 7 brainsci-15-00147-t007:** PCL-5 results with tests of fixed effects.

Type 3 Tests of Fixed Effects				
Effect	Num DF	Den DF	F Value	Pr > F
Group	3	17	0.75	0.5393
time	4	63	0.91	0.4617
time * Group	12	63	2.34	0.0149

**Table 8 brainsci-15-00147-t008:** PCL-5 results of the interaction between groups over time. The experimental treatment group demonstrates a lower mean PCL-5 score at the end of treatment.

Obs	Effect	Partial_eta_2
1	Group	0.007608
2	time	0.012400
3	time * Group	0.090299

**Table 9 brainsci-15-00147-t009:** The effect on the PCL-5 was limited and at 12 weeks post-intervention the results show a return toward higher PCL-5 mean values.

Obs	Effect	Time	Group	_time	_Group	Estimate	StdErr	DF	tValue	Probt
11	time * Group	END	A	END	B	−17.8333	7.5941	63	−2.35	0.0220
12	time * Group	END	A	END	C	−18.9167	8.4905	63	−2.23	0.0295
13	time * Group	END	A	END	D	−25.6667	8.4905	63	−3.02	0.0036
14	time * Group	END	A	END + 12	A	−23.6667	7.5941	63	−3.12	0.0028

**Table 10 brainsci-15-00147-t010:** BDI-II results for group v. time.

Type 3 Tests of Fixed Effects				
Effect	Num DF	Den DF	F Value	Pr > F
Group	3	55	2.38	0.0792
time	4	220	1.54	0.1920
time * Group	12	220	0.42	0.9540

## Data Availability

Available on request at: https://www.researchgate.net/publication/371935551_Binaural_Pulse_Modulation_BPM_as_Adjunctive_Treatment_of_AnxietyData, accessed on 29 January 2025.

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
