# Peer review of "Binaural Pulse Modulation (BPM) as an Adjunctive Treatment for Anxiety: A Pilot Study"

_brainsci, 2025, doi:10.3390/brainsci15020147_

Round 1

Reviewer 1 Report (Previous Reviewer 1)

Comments and Suggestions for Authors

The abstract should be revised to provide a clearer and more concise summary of the study, including the key objectives, methods, results, and conclusions. It currently lacks clarity in presenting the significance of the findings and the practical implications of the study.

The discussion of anxiety in psychiatric disorders requires further elaboration. Specifically, the paper should detail the nuances of anxiety's manifestation and treatment challenges across different psychiatric conditions. This would enhance the contextual relevance and depth of the study.

A dedicated subsection addressing the study’s limitations and potential directions for future research must be added. This should include limitations related to sample size, methodological constraints, and generalizability. Furthermore, recommendations for subsequent studies, such as the examination of long-term efficacy and demographic variability, should be proposed.

The methodology section should incorporate additional tests to strengthen the reliability and validity of the findings. Recommended tests include:

Hamilton Anxiety Rating Scale (HAM-A) for anxiety assessment.

State-Trait Anxiety Inventory (STAI) to differentiate between state and trait anxiety.

Physiological measures such as heart rate variability to provide objective corroboration of self-reported data.

The inclusion of G*Power analysis is necessary to justify the sample size. A detailed explanation of how the participant count was determined using this analysis should be provided within the methodology.

The discussion section would benefit from a comparative table summarizing key findings from the literature. This table should outline the similarities and differences in results across related studies, highlighting this study’s unique contributions.

A glossary or table of abbreviations should be included for all acronyms and technical terms used in the manuscript, ensuring clarity for readers.

The methods section requires critical evaluation to address potential weaknesses. Specifically:

The choice of self-reported measures as primary outcomes may introduce bias. Alternative or complementary objective measures should be considered.

The absence of detailed calibration processes for the binaural pulse device may affect replicability. Further explanation is necessary.

No physiological or neural data are presented to validate the claimed effects of BPM on neural activity, reducing the robustness of the findings.

The revised manuscript should also critically assess the potential placebo effect in the BPM treatment group and elaborate on how the control conditions were designed to minimize such biases.

Finally, a more detailed statistical analysis and reporting are required. This includes effect size for all reported tests and a deeper exploration of interaction effects in the results section.

Comments on the Quality of English Language

 The English could be improved to more clearly express the research.

Author Response

Comment 1: The abstract should be revised to provide a clearer and more concise summary of the study, including the key objectives, methods, results, and conclusions. It currently lacks clarity in presenting the significance of the findings and the practical implications of the study.

Response 1: The abstract has been revised for greater clarity.

Comment 2: The discussion of anxiety in psychiatric disorders requires further elaboration. Specifically, the paper should detail the nuances of anxiety's manifestation and treatment challenges across different psychiatric conditions. This would enhance the contextual relevance and depth of the study.

Response 2: A discussion of the nuances of anxiety in psychiatric disorders has been added to the introductory section and may be found on lines 86-104 (lines 103-121 in the marked copy).

Comment 3: A dedicated subsection addressing the study’s limitations and potential directions for future research must be added. This should include limitations related to sample size, methodological constraints, and generalizability. Furthermore, recommendations for subsequent studies, such as the examination of long-term efficacy and demographic variability, should be proposed.

Response 3: A dedicated subsection has now been added to the text as the reviewer suggests and can be found in the marked copy between lines 47-490 (lines 536-550 in the marked copy).

Comment 4: The methodology section should incorporate additional tests to strengthen the reliability and validity of the findings. Recommended tests include: Hamilton Anxiety Rating Scale (HAM-A) for anxiety assessment. State-Trait Anxiety Inventory (STAI) to differentiate between state and trait anxiety. Physiological measures such as heart rate variability to provide objective corroboration of self-reported data. The inclusion of G*Power analysis is necessary to justify the sample size. A detailed explanation of how the participant count was determined using this analysis should be provided within the methodology.

Response 4: The study, having been completed, does not allow us to retrospectively retest the participants in this pilot. The reviewer’s suggestions are appropriate and have been incorporated into the now-added limitations section of the paper (lines 473-487; in lines 532-546 in the marked copy).

Comment 5: The discussion section would benefit from a comparative table summarizing key findings from the literature. This table should outline the similarities and differences in results across related studies, highlighting this study’s unique contributions.

Response 5:  We agree that a table would be of interest. However, a well-designed meta-analysis has evaluated many aspects of BPM in reducing anxiety. Twenty-two studies were included, all showing positive effects, and a reference to that study has been included in the discussion section between lines 446-449 (lines 504-507 in the marked copy).

Comment 6: A glossary or table of abbreviations should be included for all acronyms and technical terms used in the manuscript, ensuring clarity for readers.

 Response 6: This has now been provided and may be found between lines 60-76 (lines 71-88 in the marked copy).

Comment 7: The methods section requires critical evaluation to address potential weaknesses. Specifically: The choice of self-reported measures as primary outcomes may introduce bias. Alternative or complementary objective measures should be considered.

Response 7: Again, the reviewer is correct in pointing out that self-report measures can introduce bias, and the use of physiological measurements will be included in subsequent studies, considering that our laboratories are clinical electrophysiology facilities. The instant pilot study was not funded to provide that type of measurement, and the function of the study was to determine if there was a significant effect of BPM to justify a larger scale funded investigation. This point has been incorporated in the the study limitation section (lines 476-490; lines 536-550 in the marked copy). Additonally, please see response 9 below.

Comment  8: The absence of detailed calibration processes for the binaural pulse device may affect replicability. Further explanation is necessary.

Response 8: The therapeutic effect seems to lie precisely in the psychophysical self-calibration issue. It is the fact that each individual adjusts the tones appropriate for them. As stated, “The participants were instructed to adjust the frequency control knob until they felt a slight intensification of the feeling of relaxation. Then the participants continued to focus on the feeling while they slowly adjusted the disruptor control knob until they felt an even stronger increase in the feeling or at least did not reduce the feeling.”

Comment 9: No physiological or neural data are presented to validate the claimed effects of BPM on neural activity, reducing the robustness of the findings.

Response 9: Correct, but we did refer to the recently published paper on the physiological effects of BPM. The link is attached he(re for the reviewer’s convenience. (https://link.springer.com/content/pdf/10.1186/s13256-024-04888-3.pdf) (Leisman G, Wallach J, Machado-Ferrer Y, Chinchilla-Acosta M, Meyer AG, Lebovits R, Donkin S. The effect of binaural pulse modulation (BPM) on brain state in depression and anxiety: a case series. Journal of Medical Case Reports. 2024 Nov 28;18(1):574.)

Comment 10: The revised manuscript should also critically assess the potential placebo effect in the BPM treatment group and elaborate on how the control conditions were designed to minimize such biases.

Response 10: None of the participants knew whether they were receiving sham or BPM treatment. Sham treatment had participants presented with white noise. While they were exposed to the white noise, they were instructed to engage in a cognitively demanding concentration task, such as reading or looking for details in images provided, during which they were timed at how long they were able to maintain their concentration. The experimental groups underwent the same treatment except for the self-adjusting tones. Neither group knew to which group they were assigned, nor did the investigators.

Reviewer 2 Report (Previous Reviewer 2)

Comments and Suggestions for Authors

Thank you for your comprehensive response to the initial review of the paper. The paper is significantly improved.

My main concern centres on the issue of PTSD and the use of the PCL-5. I understand that the subjects were also part of a broader study and that it was important to identify whether they had PTSD or not, and those with PTSD were excluded from this study. However, the abstract states, ' BPM was similarly effective to standard treatment approaches for anxiety, PTSD and stress.' If PTSD is excluded and, as indicated in the response (Number 5), the subjects did not meet the requirements for that diagnosis, I am concerned that the authors comment on the effectiveness of the treatment of PTSD.

Would it not be preferable to exclude the PCL score and, as per your response, state that based on DSM 5 criteria, none of the subjects had PTSD?

A review of the PCL-5 scale indicates the first eight questions all refer to an event. Given none of the subjects experienced a trauma that qualifies for criteria A, they should all score zero for these questions. Questions 9 to 20 may be considered as able to be scored, but the instructions to answer the questions state, 'Keeping your worst event in mind, please read each problem carefully and then select one of the numbers to the right  to indicate how much you have been bothered by that problem in the past month.' Therefore, in the absence of a traumatic event, the scores should be zero. It might be helpful to report the mean PCL-5 score and then explain how this score was achieved in the absence of trauma or, as suggested, delete this aspect from the paper, particularly as the paper is about anxiety, excluding PTSD.

Thank you for advising that you used Cohen's d effect size; given the small sample size, was Hedges' g used to reduce bias?  

A minor point is that some abbreviations are used before an initial elaboration in the text. The authors should recheck the text and abstract.

Author Response

Comment 1: Thank you for your comprehensive response to the initial review of the paper. The paper is significantly improved.

Response 1: You are welcome.

Comment 2: My main concern centres on the issue of PTSD and the use of the PCL-5. I understand that the subjects were also part of a broader study and that it was important to identify whether they had PTSD or not, and those with PTSD were excluded from this study. However, the abstract states, ' BPM was similarly effective to standard treatment approaches for anxiety, PTSD and stress.' If PTSD is excluded and, as indicated in the response (Number 5), the subjects did not meet the requirements for that diagnosis, I am concerned that the authors comment on the effectiveness of the treatment of PTSD.

Response 2: Thanks for pointing out the issue with PTSD. That was an error, and although the tests were administered, the data is not included in this study as it obfuscates the point. This has been removed.

Comment 3: Would it not be preferable to exclude the PCL score and, as per your response, state that based on DSM 5 criteria, none of the subjects had PTSD?

Response 3: We have done precisely this and thank you again for pointing that out.

Comment 4: A review of the PCL-5 scale indicates the first eight questions all refer to an event. Given none of the subjects experienced a trauma that qualifies for criteria A, they should all score zero for these questions. Questions 9 to 20 may be considered as able to be scored, but the instructions to answer the questions state, 'Keeping your worst event in mind, please read each problem carefully and then select one of the numbers to the right  to indicate how much you have been bothered by that problem in the past month.' Therefore, in the absence of a traumatic event, the scores should be zero. It might be helpful to report the mean PCL-5 score and then explain how this score was achieved in the absence of trauma or, as suggested, delete this aspect from the paper, particularly as the paper is about anxiety, excluding PTSD.

Response 4: Response 3 should handle the issue.

Comment 5: Thank you for advising that you used Cohen's d effect size; given the small sample size, was Hedges' g used to reduce bias?

Response 5: Yes it was.

Comment 6: A minor point is that some abbreviations are used before an initial elaboration in the text. The authors should recheck the text and abstract.

 Response 6: This has been addressed, and thank you for pointing it out.

Reviewer 3 Report (New Reviewer)

Comments and Suggestions for Authors

Abstract:
The title of the article is: “Binaural Pulse Modulation (BPM) as Adjunctive Treatment for Anxiety: A Pilot Study”, but the stated objective, “Our objective in this pilot study was to examine the possibility of using a Binaural Pulse-Mode-Modulation-type (BPM) device to restore an effective psycho-emotional state by activating the endogenous methods of self-regulation to activate the processes of recovery from anxiety and mood disorders. We desired to evaluate if emotional distress would be altered (reduced) or regulated by means of BPM-type systems”, is not fully aligned with the title. The objective addresses both anxiety and mood disorders, which creates a lack of clarity and alignment.

As a result, the abstract does not present a clear formulation of the problem or the specific objectives of the study, making the initial comprehension difficult. Additionally, the results in the abstract are overly generic and fail to highlight the key findings. For instance, “These findings indicate that over the four-week intervention period, BPM was similarly effective to standard treatment approaches for anxiety, PTSD, and stress” does not provide specific details about how BPM was effective.

Introduction:

  • The transition between the identified problem and the study objectives needs to be better structured, with a revision of the objective to align it more closely with the article’s title: “Our objective in this pilot study was to examine the possibility of using a BPM-type device to restore an effective psycho-emotional state by activating the endogenous methods of self-regulation to activate the processes of recovery from anxiety and mood disorders. We desired to evaluate if emotional distress would be altered (reduced) or regulated by means of BPM-type systems.”
  • One of the central topics in the title, “anxiety,” needs to be conceptually better defined.
  • Greater emphasis should be placed on the relevance and originality of BPM, supported by references to current studies.

Methodology:

  • The method for selecting the sample is unclear. Was it a convenience sample? Were participants from a specific hospital? Was it a randomised sample?
  • Does not specify the date of application of the study.
  • The description of the BPM method is insufficiently detailed to ensure reproducibility. What does the BPM method entail? Figure 1 is too dispersed, occupies excessive space in the article, and has a suboptimal layout. It is suggested to either describe the method in text form or significantly improve the figure.
  • There is no subsection dedicated to statistical analysis. The choice of statistical tests is not explained, and the methodology for handling missing data is not addressed.

Results:

  • The article presents numerous tables of results, but these are not accompanied by sufficient interpretative descriptions in the text.
  • The section does not adequately detail the clinical outcomes and fails to robustly address differences between groups.

Discussion:

  • The limitations of the study are not discussed in sufficient detail, and strategies to address these limitations are not proposed.

Conclusion:

  • The conclusion should include specific recommendations for future research and practical applications of the study, presented in greater detail.

Author Response

Abstract:
Comment 1: The title of the article is: “Binaural Pulse Modulation (BPM) as Adjunctive Treatment for Anxiety: A Pilot Study”, but the stated objective, “Our objective in this pilot study was to examine the possibility of using a Binaural Pulse-Mode-Modulation-type (BPM) device to restore an effective psycho-emotional state by activating the endogenous methods of self-regulation to activate the processes of recovery from anxiety and mood disorders. We desired to evaluate if emotional distress would be altered (reduced) or regulated by means of BPM-type systems”, is not fully aligned with the title. The objective addresses both anxiety and mood disorders, which creates a lack of clarity and alignment. As a result, the abstract does not present a clear formulation of the problem or the specific objectives of the study, making the initial comprehension difficult.

Response 1: We thank the reviewer for pointing out this issue and now both the background and objectives have been changed to better align with the title of the study. The changes may be found in the marked copy between lines 25-39. The concentration on anxiet has also been addressed throughout the paper.

Comment 2: Additionally, the results in the abstract are overly generic and fail to highlight the key findings. For instance, “These findings indicate that over the four-week intervention period, BPM was similarly effective to standard treatment approaches for anxiety, PTSD, and stress” does not provide specific details about how BPM was effective.

Response 2: The abstract has been revised to be less generic in its response section and noted in the marked copy of the manuscript between lines 58-63.

Introduction

Comment 3: The transition between the identified problem and the study objectives needs to be better structured, with a revision of the objective to align it more closely with the article’s title: “Our objective in this pilot study was to examine the possibility of using a BPM-type device to restore an effective psycho-emotional state by activating the endogenous methods of self-regulation to activate the processes of recovery from anxiety and mood disorders. We desired to evaluate if emotional distress would be altered (reduced) or regulated by means of BPM-type systems.” One of the central topics in the title, “anxiety,” needs to be conceptually better defined.

Response 3: We have restructured the introductory section and addressed the reviewer’s concerns. In so doing, we have added material better defining anxiety and have removed matters not directly related to what was measured. Hence, PTSD, mood disorders, etc. have been removed from the paper. The additions may be found in the marked copy between lines 106-124.

Comment 4: Greater emphasis should be placed on the relevance and originality of BPM, supported by references to current studies.

Response 4: Reference to a significant meta-analaysis of 22 relatively recent studies of BPM in reducing anxiety has been added to the discuss section and references the origninality with supportive refertences (lines 524-527 in the marked copy).

Methodology:

Comment 5: The method for selecting the sample is unclear. Was it a convenience sample? Were participants from a specific hospital? Was it a randomized sample?

Response 5:  The participants were each randomly selected from among a pool of patients from the Psychiatry outpatient clinic of the Institute for Neurology and Neurosurgery of Havana (INN) (lines 227-229 in the marked copy). The inclusion/exclusion criteria and selection and randomization procedures are specified between lines 253-290 in the marked copy.

Comment 6: Does not specify the date of application of the study.

Response 6: The study was approved by the INN research review board on 7 February 2022 and commenced immediately thereafter. A statement to that effect appears in the marked copy between lines 243-248.

Comment 7: The description of the BPM method is insufficiently detailed to ensure reproducibility. What does the BPM method entail? Figure 1 is too dispersed, occupies excessive space in the article, and has a suboptimal layout. It is suggested to either describe the method in text form or significantly improve the figure.

Response 7: We would love to remove the figure; however, a third reviewer insisted that we produce a flow chart of the experimental procedure and this we have included in the response. The resolution will be improved if and when the paper is accepted for publication. Save that, we have rewritten the BPM procedure now appearing between lines 320-337 in the marked copy.

Comment 8: There is no subsection dedicated to statistical analysis. The choice of statistical tests is not explained, and the methodology for handling missing data is not addressed.

Response 8: There is no missing data. It is one of the benefits of doing clinical research in Cuba, where patients are strongly motivated to participate. A subsection dedicated to statistical analysis has now been added and can be found in the marked copy between lines 321-332.

Results:

Comment 9: The article presents numerous tables of results, but these are not accompanied by sufficient interpretative descriptions in the text. The section does not adequately detail the clinical outcomes and fails to robustly address differences between groups.

Response 9: We have added in the results section comments about the robust statistical effects and that a decline in those effects were noted when retesting 12 post-intervention.

Discussion:

Comment 10: The limitations of the study are not discussed in sufficient detail, and strategies to address these limitations are not proposed.

Response 10:  We agree with the reviewer that the study’s limitations have not been adequately discussed. As such, we have, in the discussion section, included a separately headed section on the study limitations. This material may be found between lines 556-570.

Conclusion:

Comment 11: The conclusion should include specific recommendations for future research and practical applications of the study, presented in greater detail.

Response 11: We have added recommendations for future research and practical applications to the conclusions section (lines 573-590 in the marked copy).

Round 2

Reviewer 1 Report (Previous Reviewer 1)

Comments and Suggestions for Authors

The authors have completely addressed all my comments, and I have no further concerns. Therefore, I recommend accepting the paper.

Author Response

Thanks you for the recommendation to accept

Reviewer 3 Report (New Reviewer)

Comments and Suggestions for Authors

Most of the references cited by the authors are over five years old, making the bibliography significantly outdated.

The "Flow Chart of BPM Administration" is poorly formatted as an image. It occupies excessive space in the article and contains asymmetrical boxes. It is suggested to either explain the information in text form or improve the image's layout.

There are too many consecutive tables without introductory text before each one. Including an explanatory text preceding each table would enhance clarity and readability.

Author Response

1. Most of the references cited by the authors are over five years old, making the bibliography significantly outdated.

1. The reference list has been updated and made significantly more relevant. Duplicates have been removed as well

2. The "Flow Chart of BPM Administration" is poorly formatted as an image. It occupies excessive space in the article and contains asymmetrical boxes. It is suggested to either explain the information in text form or improve the image's layout.

2. The flow chart was required by another reviewer and the picture will be replaced by text prior to the production office involvement. This issue has painstakingly been resolved even with the internet down in Cuba. The flow chart occupies a single page.

3. There are too many consecutive tables without introductory text before each one. Including an explanatory text preceding each table would enhance clarity and readability.

We have addressed this issue.

This manuscript is a resubmission of an earlier submission. The following is a list of the peer review reports and author responses from that submission.

Round 1

Reviewer 1 Report

Comments and Suggestions for Authors

I have reviewed the article titled "Binaural Pulse Modulation (BPM) as Adjunctive Treatment for Anxiety," which explores the efficacy of BPM in treating anxiety, PTSD, and stress through a sound-based modulation technique. The study presents a novel therapeutic approach but requires significant revisions for clarity, rigor, and a more robust methodology.

The paper would benefit from an ablation study to isolate and verify the specific effects of BPM on anxiety reduction. Such a study would help in identifying which aspects of the BPM protocol contribute most to the observed effects, allowing for a clearer understanding of its therapeutic mechanisms.

A more explicit discussion of the study's limitations, especially regarding the short-term effects of BPM and the lack of long-term follow-up, is recommended. Additionally, outlining directions for future research, such as optimizing the frequency parameters for BPM and evaluating its applicability across different demographics, would enrich the discussion.

To improve readability, consistent use and definition of technical terms and abbreviations, such as BPM and EEG, should be applied throughout the text.

The methodology section would be enhanced by a flowchart or block diagram showing the experimental setup and BPM calibration steps. This would clarify the study design and procedures, especially the use of individual tuning for BPM frequency settings.

Expanding the introduction to explicitly state the research gap and how BPM distinguishes itself from similar sound-based interventions would add relevance. Furthermore, the discussion should compare BPM's results to those of traditional anxiety treatments more comprehensively, demonstrating where BPM shows promise as an adjunctive or alternative therapy.

Providing access to the study data or BPM device settings would improve reproducibility and transparency, facilitating further research and potential clinical application.

The results section could benefit from more detailed data on parameter optimization and individual variability in BPM response. Additionally, the authors should elaborate on the temporal limitations of BPM's effects, as indicated by the return to baseline scores in post-treatment follow-ups.

Clarifying the novelty of BPM compared to existing auditory interventions, specifically the unique effects of the dual frequency modulation on mood regulation, would strengthen the paper's contribution to the field.

Improved commentary on the statistical analysis and interpretation of figures and tables, particularly those showing time-course data and effect size across groups, would help readers understand the study's outcomes.

Finally, for further insight into auditory and cognitive modulation, the authors may wish to reference the article "A novel ternary pattern-based automatic psychiatric disorders classification using ECG signals" from recent literature, which could provide context for exploring BPM’s applications in mental health interventions.

Addressing these points will strengthen the study’s validity, improve its impact, and clarify its potential as a non-invasive adjunctive therapy for anxiety and related disorders.

Comments on the Quality of English Language

The English could be improved to more clearly express the research.

Author Response

Comment 1: I have reviewed the article titled "Binaural Pulse Modulation (BPM) as Adjunctive Treatment for Anxiety," which explores the efficacy of BPM in treating anxiety, PTSD, and stress through a sound-based modulation technique. The study presents a novel therapeutic approach but requires significant revisions for clarity, rigor, and a more robust methodology.

The paper would benefit from an ablation study to isolate and verify the specific effects of BPM on anxiety reduction. Such a study would help in identifying which aspects of the BPM protocol contribute most to the observed effects, allowing for a clearer understanding of its therapeutic mechanisms.

Response 1: We greatly agree with the reviewer’s suggestion, but given the nature of our laboratory’s principal thrust in movement and cognition and resonance effects on cognition in humans developmentally, we can certainly suggest that this was one of the limitations of the study and that others are cordially invited to investigate this aspect of the work. However, we have made reference to the published work of others that may be relevant in this context. [That may be found in the marked copy of the discussion section between lines 328-368]

Comment 2: A more explicit discussion of the study's limitations, especially regarding the short-term effects of BPM and the lack of long-term follow-up, is recommended. Additionally, outlining directions for future research, such as optimizing the frequency parameters for BPM and evaluating its applicability across different demographics, would enrich the discussion.

To improve readability, consistent use and definition of technical terms and abbreviations, such as BPM and EEG, should be applied throughout the text.

Response 2: We value the reviewer's suggestion beyond platitudes and have added to the paper significantly in its discussion section by expanding the comments pertaining to the results reflecting short-term effects with long-term follow-up effects not having been investigated beyond the multiple testings. Opportunity and funding were not available at this juncture, and that is likely to change. In the interim, it is a significant limitation of the study. We have thusly indicated so in the discussion section reflected in the marked copy between lines 397-402.

Besides noting that long-term follow-up is necessary and, therefore, a limitation, we have additionally addressed the issue by changing the title of the paper to, “Binaural Pulse Modulation (BPM) as Adjunctive Treatment for Anxiety: A Pilot Study”. The paper was not intended as a clinical trial and, as such was not registered with the FDA. A ststatement to that effect appears in the marked revised manuscript  between lines 194-195. A long-term efficacy study would certainly be the next order of business.

Additonally, The use of acronyms and technical terms have been made consistent throughout the text.

Comment 3: The methodology section would be enhanced by a flowchart or block diagram showing the experimental setup and BPM calibration steps. This would clarify the study design and procedures, especially the use of individual tuning for BPM frequency settings.

Response 3: A block diagram has now been provided.

Comment 4: Expanding the introduction to explicitly state the research gap and how BPM distinguishes itself from similar sound-based interventions would add relevance. Furthermore, the discussion should compare BPM's results to those of traditional anxiety treatments more comprehensively, demonstrating where BPM shows promise as an adjunctive or alternative therapy.

Response 4: We have added to the introductory section to reflect the “research gap” and how the instant technology differs from similar devices currently applied. We are actually less interested in the singular device and more in how it is that resonance in general and aural feedback in particular can affect anxiety and other behavioral issues. Furthermore, we have referenced a well written  a comparative analysis the introducotry section where the reader can examine the differences between auditory monaural and binaural beats in a therapeutic context and how BPM differs from other approaches (both a similar and non-similar) and the possible reasons why there is an effect. We have attempted to do so without “hand waving.”[lines120-128 in marked copy]

Comment 5: Providing access to the study data or BPM device settings would improve reproducibility and transparency, facilitating further research and potential clinical application.

Response 5: The data has always been available and is indicated in the paper itself. For the reviewer’s benefit, it may be found at the following website: (https://www.researchgate.net/publication/371935551_Binaural_Pulse_Modulation_BPM_as_Adjunctive_Treatment_of_AnxietyData)

Comment 6: The results section could benefit from more detailed data on parameter optimization and individual variability in BPM response. Additionally, the authors should elaborate on the temporal limitations of BPM's effects, as indicated by the return to baseline scores in post-treatment follow-ups.

Response 6: We thank the reviewer for pointing this out and have now significantly expanded both the procedure and results sections.

Comment 7: Clarifying the novelty of BPM compared to existing auditory interventions, specifically the unique effects of the dual frequency modulation on mood regulation, would strengthen the paper's contribution to the field.

Response 7: This, we think, has been significantly enhanced by expanding the discussion section and highlighting the differences between existing auditory interventions and this form of pulse modulation for the treatment of anxiety.

Comment 8: Improved commentary on the statistical analysis and interpretation of figures and tables, particularly those showing time-course data and effect size across groups, would help readers understand the study's outcomes.

Response 8: Adjustments have been made based on the reviewer’s suggestions

Comment 9: Finally, for further insight into auditory and cognitive modulation, the authors may wish to reference the article "A novel ternary pattern-based automatic psychiatric disorders classification using ECG signals" from recent literature, which could provide context for exploring BPM’s applications in mental health interventions.

Response 9: Done

Comment 10: Addressing these points will strengthen the study’s validity, improve its impact, and clarify its potential as a non-invasive adjunctive therapy for anxiety and related disorders.

Reviewer 2 Report

Comments and Suggestions for Authors

Thank you for the opportunity to review the paper.

I will address the issues I have noted in point format.

1) The abstract indicates the BPM group was significantly better than the sham treatment. The abstract should include the significance value. I was unable to locate in the abstract a description of the sham group or the sham treatment. The study also did not identify how people were allocated to either the treatment or sham group.

2) The methods section indicates each participant was diagnosed with major depression or anxiety but did not advise whether DSM 5 or ICD 11 criteria were used.

3) The paper indicates hypo/hyperthyroidism was excluded but not whether other disorders that require testing were identified and excluded from the study.

4) The paper states the participants did not have a history of trauma (psychological or physical). If that is correct, it seems unusual that each person was then required to complete a PCL-5 as detailed in the abstract.

6) The inclusion criteria indicate the participants met the criteria for an anxiety disorder but do not identify which disorder except for PTSD. Again, it seems unusual that a group with no history of trauma could meet the criteria for PTSD (None of the individuals examined presented with a history of neurological disease or disorder, history of seizures, or trauma either psychological or physical).

7) The study criteria indicate that none of the participants suffered from a primary psychiatric complaint that was a non-anxiety disorder, yet the procedure identifies depression and anxiety but not how it was determined which was the primary disorder.

8) Among the self-report inventories used in the study is the Geriatric Depression Scale; does this mean some subjects were over 65? Nor is evidence provided as to the duration of the symptoms. The age range should be included in the paper.

9) The results section reports differences between groups, e.g. GAD 7, but it is unclear between which groups. The tables use abbreviations eg Num DF and Den DF, without an explanation of the meaning of the abbreviations. 

10) There is a reference to the effect size, is this Cohens or some other measure?

11) Can the authors clarify the information in Table 4? It reports on the experimental group and all other groups, what were these other groups? Again, there are abbreviations without explanation (Table 4  Interaction between the experimental treatment group compared to all the other groups at the end of treatment, with the experimental treatment group having a lower mean GAD-7 score at  ..... ).

12) Table 7 reports on the impact of the treatment on the PCL 5 scores, aside from the absence of data on the PCL scores it is unclear why it should be assessed when by definition none of the patient groups could have PTSD. 

13) Given the gender split of the participants, it would be useful for a comparison between males and females in the treatment group, and it would be helpful to have gender data pertaining to the non-treatment and sham treatment groups.

14) It would be helpful for the authors to identify the strengths and limitations of the study.